# Correction of Beta-Thalassemia IVS-II-654 Mutation in a Mouse Model Using Prime Editing

**DOI:** 10.3390/ijms23115948

**Published:** 2022-05-25

**Authors:** Haokun Zhang, Ruilin Sun, Jian Fei, Hongyan Chen, Daru Lu

**Affiliations:** 1State Key Laboratory of Genetic Engineering, MOE Engineering Research Center of Gene Technology, School of Life Sciences, Fudan University, Shanghai 200438, China; zhanghaokun666@hotmail.com; 2Shanghai Model Organisms Center, No.3577 Jinke Rd., Shanghai 201203, China; ruilin.sun@modelorg.com (R.S.); jian.fei@modelorg.com (J.F.); 3NHC Key Laboratory of Birth Defects and Reproductive Health, Chongqing Key Laboratory of Birth Defects and Reproductive Health, Chongqing Population and Family Planning, Science and Technology Research Institute, Chongqing 404100, China

**Keywords:** prime editing, beta-thalassemia, IVS-II-654 mutation, genome editing, gene therapy

## Abstract

Prime editing was used to insert and correct various pathogenic mutations except for beta-thalassemia variants, which disrupt functional beta-globin and prevent hemoglobin assembly in erythrocytes. This study investigated the effect of gene correction using prime editor version 3 (PE3) in a mouse model with the human beta-thalassemia IVS-II-654 mutation (C > T). The T conversion generates a 5′ donor site at intron 2 of the beta-globin gene resulting in aberrant splicing of pre-mRNA, which affects beta-globin expression. We microinjected PE3 components (pegRNA, nick sgRNA, and PE2 mRNA) into the zygotes from IVS-II-654 mice to generate mutation-edited mice. Genome sequencing of the IVS-II-654 site showed that PE3 installed the correction (T > C), with an editing efficiency of 14.29%. Reverse transcription-PCR analysis showed that the PE3-induced conversion restored normal splicing of beta-globin mRNA. Subsequent comprehensive phenotypic analysis of thalassemia symptoms, including anemic hematological parameters, anisocytosis, splenomegaly, cardiac hypertrophy, extramedullary hematopoiesis, and iron overload, showed that the corrected IVS-II-654 mice had a normal phenotype identical to the wild type mice. Off-target analysis of pegRNA and nick sgRNA additionally showed the genomic safety of PE3. These results suggest that correction of beta-thalassemia mutation by PE3 may be a straightforward therapeutic strategy for this disease.

## 1. Introduction

Beta-thalassemia is a hematopoietic disease caused by a single-gene mutation of the hemoglobin (HGB) subunit beta gene (*HBB*), leading to anomalous beta-globin expression [1]. According to the World Health Organization, beta-thalassemia is a prevalent genetic disease, with over 90 million carriers in the world and over 40,000 children born with various alleles every year [2]. More than 300 beta-thalassemia alleles have been identified to date, with point mutations in introns being the most common type, leading to activation of abnormal splicing sites [1,3]. In the IVS-II-654 mutation (C > T) for example, the T conversion at nucleotide (nt) 654 of intron 2 generates an additional 5′ donor splice site at 652 and activates an endogenous recessive 3′ acceptor site at 579 (Figure 1a) [4,5]. Accordingly, the spliced IVS-II-654 transcript contains nucleotides 580-652 of the second intron, which newly generates a premature stop codon, and cannot encode a functional beta-globin protein [4,5,6]. This mutation is prevalent among patients in East and Southeast Asia, accounting for 20% of beta-thalassemia cases in some regions [6,7].

In 1998, Lewis et al. used the “plug and socket” method of gene targeting to replace the two normal murine adult beta-globin genes with a single copy of the human IVS-II-654 allele [5]. Heterozygous *Hbb^th−4^*/*Hbb*^+^ mice (designated as “beta” in this study) showed the same abnormal splicing of *HBB* mRNA and exhibited pathological thalassemia phenotypes, including anisocytosis, splenomegaly, and visceral iron overload [5]. This model has since been used in many studies of beta-thalassemia treatments. For example, RNA interference-induced knockdown of alpha-globin mRNA, combined with antisense RNA to reduce abnormal beta-globin mRNA, efficiently ameliorated thalassemia symptoms in beta mice [8,9,10]. Lentiviral vector-induced re-expression of the normal human *HBB* gene also successfully resolved anemic complications [11,12]. Fang et al. used transcription activator-like effector nucleases to induce a non-homologous end joining (NHEJ) near the IVS-II-654 mutation site, effectively restoring *HBB* mRNA splicing by destroying the 5′ donor splice site at nt 652 of intron 2, thus curing the symptoms in beta mice [13]. In addition, Lu et al. performed CRISPR/Cas9-mediated genome editing, targeting both the IVS-II-654 (C > T) and the 3′ acceptor site at 579, and corrected abnormal beta-globin mRNA splicing in beta mice [14].

The current study aimed to correct the IVS-II-654 mutation in beta mice using prime editing. The prime editor (PE) is a recently developed genome-editing tool that introduces base-pair conversions, small insertions, and small deletions in a controlled manner [15,16]. The PE protein is combined with a Cas9 nickase (nCas) domain and a reverse transcriptase (RT) domain, which enable the PE protein to target the editing site using an engineered prime editing guide RNA (pegRNA), and encodes the intended DNA strand according to the RT template at the 3′ end of the pegRNA, thus installing the desired edits. The RT template usually contains the immediate regions homologous to the target site for improving DNA repair. Version 3 of the PE system (PE3) is the most commonly used version, and adds sgRNA targeting the non-edited strand for nicking, further increasing the editing efficiency [15].

As shown in Figure 1b, by microinjection of PE3 RNAs, the zygotes from female beta mice develop into edited embryos with a corrected IVS-II-654 mutation, which in turn develop into mice with a normal phenotype. The pegRNA of PE3 targets the IVS-II-654 mutation site with a therapeutic RT-template at the 3′ end (Figure 1c, Appendix A). The RT-template of pegRNA was designed to induce a GGT > CCC conversion at the IVS-II-654 site (Figure 1c), which repairs the beta-thalassemia mutation and edits the protospacer adjacent motif (PAM) region for *Streptococcus pyogenes* (Sp) Cas9 to prevent secondary editing. The nCas9 domain of PE protein targets the mutation site according to the space sequence of pegRNA, and the RT domain subsequently encodes the therapeutic DNA strand with intended GGT > CCC conversion at the IVS-II-654 site. Finally, the pathogenic C > T mutation at the IVS-II-654 site is replaced with a therapeutic conversion of CCC at the nucleotides 652-654 of the second intron, which prevents abnormal splicing of pre-mRNA due to the IVS-II-654 mutation. We also carried out molecular genetic and phenotypic examinations of the edited pups to test the editing efficiency of PE3 at the IVS-II-654 site, and investigated the effects of the PE3-induced gene treatment in terms of resolving the beta-thalassemia symptoms in beta mice.

## 2. Results

### 2.1. Genotyping Analysis Showed That PE3 Installed Desired On-Target Edits at the IVS-II-654 Site

We performed PE3-induced correction of the IVS-II-654 mutation by microinjecting the PE2 RNA (coding for nCas9 domain and RT domain), pegRNA, and nick sgRNA into murine zygotes from female beta mice (Figure 1b). The injected zygotes were then transplanted into pseudopregnant female mice, which produced 28 surviving pups. We identified five beta-genotype and 23 wild type (WT) pups, according to the Jackson Laboratory protocol for genotyping of *Hbb^th−4^*/*Hbb*^+^ (beta) and *Hbb*^+^/*Hbb*^+^ (WT) (data not shown).

We then performed Sanger sequencing to determine the editing outcomes at the IVS-II-654 site. The sequencing traces showed PE3 installed editing of GGT > CCC in beta-genotype pups (designated beta-PE, Table 1) and GGC > CCC in WT pups (designated WT-PE) (Table 1, Figure 2a and Appendix A). We distinguished between these two edits as “correction” and “modification”, respectively, according to their therapeutic effect. In WT pups, PE3 induced the CCC conversion near the target site (designated WT-PE with by-product, WT-PEB) and generated by-product edits at the nick sgRNA site (designated WT-nick) (Table 1, Figure 2a and Appendix A). PE3 eventually installed the intended CCC conversion at the IVS-II-654 site in four mice (efficiency: 14.29%) and generated by-product edits in nine mice (efficiency: 32.14%).

The IVS-II-654 mutation causes abnormal splicing of *HBB* mRNA, leading to an intronic 73-nt sequence insertion and accordingly making the mutant mRNA longer than the WT mRNA (Figure 1a). We therefore extracted RNAs and performed RT-polymerase chain reaction (PCR) analysis to determine if the PE3-induced gene correction restored normal *HBB* mRNA splicing in beta-PE mice. The *HBB* transcript in beta pups was longer than those in WT, beta-PE, WT-PE, WT-PEB, and WT-nick pups (Figure 2b). The Sanger sequencing results of RT-PCR products further indicated that the inserted sequence in the *HBB* transcript in beta pups was removed in beta-PE pups (Appendix A), indicating that the PE3-installed GGT > CCC conversion successfully repaired the pathogenic splicing of *HBB* mRNA in beta-PE mice. RT-PCR analysis also showed that the *HBB* transcripts in WT-PE, WT-PEB, and WT-nick mice were identical to WT mice, despite introducing by-product edits near pegRNA and nick sgRNA, indicating that these by-product edits did not affect *HBB* mRNA splicing in WT mice.

Notably, some beta-PE mice (17# and 18#) had coexistence of corrected “CCC” (relatively dominant) and mutant “GGT” at the IVS-II-654 site (Figure 2a and Appendix A), indicating that they were edited chimerically. However, genome sequencing of the F1 pups produced by crossing a beta-PE male (18#) with a WT female showed a clean GGT > CCC conversion at the IVS-II-654 site (Figure 2c and Appendix A), and accordingly restored *HBB* mRNA (Appendix A), indicating that the PE3-induced correction could be inherited by the next generation.

### 2.2. Phenotypic Analysis Showed That PE3-Induced Treatment Effectively Cured Thalassemia Symptoms

We examined the symptoms underlying insufficient HGB, abnormal erythropoiesis, and erythroid maturation deficiency to determine if beta-thalassemia was cured in beta-genotype mice. We obtained peripheral blood samples from WT, beta, beta-PE, and beta-PE F1 mice, and analyzed red blood cell count (RBC), HGB concentration, hematocrit (HCT), mean corpuscular volume (MCV), mean corpuscular hemoglobin (MCH), red cell distribution width_standard deviation (RDW_SD), platelet count (PLT), white blood cell count (WBC), lymphocyte count (LYM), and minimum inhibitory dilution (MID). Compared to WT, beta mice showed a significantly lower RBC, HBG, HCT, MCV, and MCH and a significantly higher RDW, PLT, WBC, LYM, and MID (Figure 3a, Appendix A). By contrast, these parameters were restored to WT levels in beta-PE and F1 mice. Wright–Giemsa-stained blood smears showed that beta mice had severe anisocytosis, poikilocytosis, and target cells, whereas beta-PE and F1 mice showed healthy erythrocytes, consistent with the blood smears in WT mice (Figure 3d). These results thus showed that PE3-induced gene correction successfully cured hematological symptoms in beta-genotype mice.

We also examined the histopathological symptoms in WT and beta mice by comparing tissue coefficients, extramedullary hematopoietic progenitor cells, and iron deposition in 10 essential tissues, including spleen, heart, brain, kidney, liver, lung, thymus, testis, epididymis, and ovary with the uterus, between WT and beta mice. Compared with WT mice, beta mice showed significant splenomegaly and cardiac hypertrophy (Appendix A, Appendix A) due to hemolysis-derived inflammatory swelling and anemia-derived compensatory hypertrophy. Ferrocyanide iron-stained tissue sections (Appendix A) showed severe iron deposition in the liver, kidney, and spleen in beta mice, with apparent extramedullary hematopoietic progenitor cells in the liver and acute disintegration of white pulp in the spleen in beta mice.

We subsequently examined the coefficients of the spleen and heart in beta-PE and beta-PE F1 mice. Thalassemia-featured splenomegaly and cardiac hypertrophy were absent in the beta-PE and F1 mice, indicating successful cure of hemolysis and anemia (Figure 3b,c, Appendix A). In addition, ferrocyanide iron-stained sections of the liver, kidney, and spleen from beta-PE and beta-PE F1 mice showed that extramedullary hematopoietic progenitor cells, iron deposition, and white pulp disintegration were also absent (Figure 3e–g), demonstrating that the thalassemia-derived extramedullary hematopoiesis, iron overload, and spleen inflammation were significantly improved.

Our results indicated that IVS-II-654 mutation-derived hematological and histopathological symptoms were ameliorated in beta-genotype mice using PE3-induced gene correction.

### 2.3. Off-Target Analysis Showed That PE3-Induced IVS-II-654 Correction Was Safe in the Human Genome

PE3 was recently shown to induce by-product edits at the position of nick sgRNA [17], consistent with our results of WT-nick (Figure 2a and Appendix A). We thus investigated the human-genome safety of the PE3-induced IVS-II-654 correction by off-target analysis of the pegRNA (Peg) and nick sgRNA (Ni).

To this end, we constructed a plasmid containing Cas9, the reporter gene enhanced green fluorescent protein (*EGFP*), and sgRNA (Figure 4a). Two plasmids targeting different sites, Peg and Ni, were transferred into human HEK293T cells. *EGFP*-positive cells were then collected by fluorescence activated cell sorting (FACS) (Figure 4a) and proliferated in vitro to extract genomic DNA for deep sequencing of the off-target edits of Peg and Ni, respectively. We searched the off-targets edits using the webtool Cas-OFFinder, and selected the three off-target sites where Peg and Ni were most likely to generate spCas9-induced indels in the human genome for deep sequencing. The overall indel-inducing efficiency at the off-target sites of Peg and Ni was <0.35% (Figure 4b, Appendix A), indicating that the PE3-mediated gene treatment strategy for the beta-thalassemia IVS-II-654 mutation had good human-genome safety.

## 3. Discussion

The current results showed that PE3 successfully installed a therapeutic conversion of GGT > CCC and accordingly ameliorated the hematological and histopathological symptoms in a beta-thalassemia mouse model with the IVS-II-654 mutation. CRISPR/Cas9-derived gene modification, base editing, and prime editing have been utilized for precise gene therapies [16,18,19,20]. CRISPR/Cas9-mediated gene modification mainly involves homology-directed repair (HDR), NHEJ, microhomology-mediated end joining (MMEJ), and homology-mediated end joining (HMEJ) [18,20,21]. We successfully corrected the IVS-II-654 (C > T) mutation in the same mouse model using a Cas9-mediated HMEJ method, and are planning to publish the results elsewhere. However, this therapeutic strategy is greatly limited by the safety issues associated with inducing indel by-products in the genome [1,22,23,24]. Base editing (BE, also for base editor) installs three main types of base-pair conversions and accordingly has three main classes, including cytosine BE (catalyzing the conversion of C/G > T/A), adenine BE (catalyzing A/T > G/C conversions), and glycosylase BE (mediating C > G substitution) [16,19,25,26]. The BE-induced therapeutic strategy is limited by the editing window and introduction of bystander mutations [18,19]. For example, although the adenine BE can install the T > C conversion, it was unsuitable for gene correction in this study because of the limitation of the editing window at the IVS-II-654 site. Cas9-mediated modification and BE have so far been used to modify the enhancer of transcription factor B-cell lymphoma/leukemia 11A (*BCL11A*), the core sequence of which is required to repress fetal hemoglobin (alpha2gamma2) in adult-stage erythroid cells to ameliorate beta-thalassemia-derived disorders [16,27,28,29,30,31,32,33,34,35]. Our results showed that PE could efficiently install therapeutic corrections in a targeted and precise manner, suggesting that future PE3-induced beta-thalassemia therapy could be used to modify the *BCL11A* enhancer and induce precise corrections at each mutation site.

However, although PE has been reported to be capable of correcting up to 89% of known pathogenic mutations associated with human diseases [15], its effect is usually finite due to the limited efficiency and specificity of pegRNA for facilitating spCas9-induced editing at the target site [18,36,37,38,39]. *Staphylococcus aureus* (Sa) Cas9 (PAM: NNGRRT) and engineered SpCas9 (PAM: NG/NHN)-based PE systems have been developed to resolve these issues and have induced on-target edits with desirable editing efficiencies [39,40]. According to the webtool CRISPOR, the prediction results showed that the spacer sequence of pegRNA in this study had medium efficiency and specificity at the IVS-II-654 site, while the counterpart of *Neisseria meningitidis* (Nme2) Cas9 (PAM: NNNNCC) had higher efficiency and specificity [41]. This suggests that future prime editing could use alternative nCas9 domains to improve the genome accessibility and target specificity of pegRNA [39,40].

This study evaluated a straightforward therapeutic strategy for beta-thalassemia by microinjecting PE3 RNAs into mouse zygotes. However, therapeutic editing of human germline cells is ethically forbidden, due to potential by-product edits in the genome [23,42,43]. The practicable therapeutic strategy for beta-thalassemia thus involves editing human-peripheral-blood-mobilized CD34^+^ hematopoietic stem and progenitor cells (HSPSc) by ribonucleoprotein (RNP) electroporation. By editing the *BCL11A* enhancer core sequence, modified HSPSc can be infused into myeloablated patients to rebuild the hematopoietic system [33,34,35]. A recent study showed that PE3 RNP electroporation could efficiently introduce intended edits in germlines and primary cells [17], suggesting that a similar method could be used to modify HSPSc in vitro for beta-thalassemia therapy. In addition, lipid nanoparticle (LNP) technologies have recently made rapid progress, and high-throughput screening of LNP formulas with different proportions of polyethylene glycol, amino (cationic), and structural lipids, and cholesterol, may lead to the selection of HSPSc-targeting formulas with high efficiency, thus enabling therapeutic applications of PE3 RNAs [44,45,46,47,48,49,50]. Compared with the strategy of in vitro editing of HSPSc, the LNP-mediated method eliminates the process of myeloablation in patients, suggesting that this strategy may be safer and more effective in patients with beta-thalassemia.

## 4. Materials and Methods

### 4.1. Animal Model and Study Approval

The beta-thalassemia IVS-II-654 mice, termed *Hbb^th−4^*/*Hbb*^+^ (stock No: 003250), were obtained from the Jackson Laboratory (Bar Harbor, ME, USA). All mice were housed in a pathogen-free facility at 25 ± 1 °C on a 12-h light/dark cycle with free food and tap water access. All animal operations conformed to the regulations drafted by the Association for Assessment and Accreditation of Laboratory Animal Care in Shanghai, and the study protocol was reviewed and approved by the Institutional Animal Care and Use Committee, Fudan University, China. All pups were numbered immediately after birth. However, the pups with the *Hbb^th−4^*/*Hbb*^+^ are usually fragile due to thalassemia mutation, and some of them will die after birth. Therefore, to protect the *Hbb^th−4^*/*Hbb*^+^ pups, the genotyping analysis of *Hbb^th−4^*/*Hbb*^+^ (beta) and *Hbb*^+^/*Hbb*^+^ (WT) was performed two weeks after birth using the tail tips-derived genomic DNA using the PCR method from the Jackson Laboratory. The PCR primers used for genotyping are listed in Appendix A.

### 4.2. Plasmid Construction

The pU6-pegRNA-GG-Vector was purchased from Addgene (Watertown, MA, USA) (132777) [15]. The pegRNA plasmid was constructed according to the reported protocol [15]. In the present study, we designed the required pegRNA and nick sgRNA for IVS-II-654 correction using the website tool pegFinder (accessed on 19 November 2020) [51]. pegRNA plasmid backbone was cut by BsaI-HFv2 (NEB, Ipswich, MA, USA) for overhangs. T7 + spacer oligos and 3′ extension oligos were synthesized (Appendix A). Phosphorylating sgRNA scaffold sequence with T4 PNK (NEB, Ipswich, MA, USA), spacer sequence, 3′ extension sequence, and sgRNA scaffold sequence were cloned into the backbone of pegRNA expression vector with T4 DNA ligase (NEB, Ipswich, MA, USA). For the construction of T7 + nick sgRNA, oligos were synthesized (Appendix A), annealed, and cloned into the BbsI site of the PSK-u6 + gRNA expression vector (Appendix A).

### 4.3. In Vitro Transcription

The pCMV-PE2 was purchased from Addgene (132775) [15]. The PE2 plasmid was linearized by the PmeI enzyme (NEB, Ipswich, MA, USA), and in vitro transcription was performed using the T7 Ultra Kit (Ambion, Austin, TX, USA). According to the manufacturer’s protocols, RNA was purified by the Mini Kit (Qiagen, Hilden, Germany). The expression plasmids of T7 + pegRNA and T7 + nick sgRNA were linearized by the XhoI enzyme (NEB, Ipswich, MA, USA). According to the manufacturer’s protocols, the pegRNAs and sgRNAs were transcribed in vitro by the MEGA shortscript Kit (Invitrogen, Carlsbad, CA, USA) and purified by the MEGA clear Kit (Invitrogen, Carlsbad, CA, USA).

### 4.4. Embryo Microinjection and Identification of Gene Editing Mice

To induce the mutation correction in the beta mouse model, 4-week-old beta females were superovulated with pregnant mare serum gonadotropin and human chorionic gonadotropin 48 h before mating to beta male. Embryos were collected from the oviducts of female mice. The microinjection was performed (Eppendorf FemtoJet^®^ 4i microinjector, Hamburg, Germany) with the mRNA mixture containing pegRNA (50 ng/µL), nick sgRNA (16.7 ng/µL), and PE2 mRNA (100 ng/µL), into the pronuclei at the single-cell stage. The injected embryos were cultured in KSOM at 37 °C, 5% CO_2_ for 2 days, and then transplanted into pseudopregnant mice for breeding. Two weeks after birth, 0.5 cm tail tips were obtained from the young mice for genomic DNA extracting and sequencing to identify the gene-editing mice.

### 4.5. RT-PCR Analysis

Total RNA was extracted using TRIzol (Thermo Fisher Scientific, Waltham, MA, USA) from mouse blood samples, followed by reverse-transcription with HiScript Q RT SuperMix for qPCR (Vazyme, Shanghai, China). The whole blood samples (100 µL) were collected from the six-week-old mice (anesthetized before blood sample collection). RT-PCR was processed with the specific primers for detecting IVS-II-654 mutant cDNA (Appendix A). The mouse *GAPDH* gene was RT-PCR amplified as an internal control. PCR was performed using KOD Neo (Toyobo, Shanghai, China) with the following conditions: initial denaturation at 98 °C for 1 min and then 35 cycles of denaturation at 98 °C for 10 s, annealing at 60 °C for 15 s, and extension at 68 °C for 1 min, followed by a final extension at 68 °C for 2 min.

### 4.6. Hematological Analysis

Before blood sample collection, the six-week-old mice were nicely anesthetized. The whole blood samples (150 µL) from the mice were collected. The routine blood tests, according to the manufacturer’s protocol, were determined using Hematology Analyzer (HA-300Vet, Biotoo, Guilin, China). The peripheral blood smears were stained with Wright-Giemsa (BASO, BA4017, Zhuhai, China). The WT group and beta group were used as negative and positive controls, respectively. The raw data of hematological analysis, including each group and total number, is listed in Appendix A.

### 4.7. Histopathology

After blood sample collection, the six-week-old mice were nicely euthanized, weighed, and dissected. The spleen, heart, brain, kidney, liver, lung, thymus, testis, epididymis, and ovary with the uterus were weighed and were used for tissue pathology analysis. The spleen and heart were imaged. Small pieces of tissue were embedded in paraffin wax and sliced into 4 μm thick sections. Iron accumulation in the liver, spleen, and kidney was determined by ferrocyanide iron staining (BestBio, BB-44371, Shanghai, China). The tissue coefficient was calculated as tissue mass divided by body mass. The WT group and beta group were used as negative and positive controls, respectively. The raw data of tissue coefficient, including each group and total number, are listed in Appendix A.

### 4.8. Statistical Analysis

Statistical analysis was performed using GraphPad Prism (version 7.00, GraphPad Software, Inc., San Diego, CA, USA). A Student’s *t*-test was used for inter-group comparisons, and *p* values < 0.01 were considered significant.

### 4.9. Cell Culture and Transfection

To analyze the off-targets referring to spacer sequences of pegRNA and nick sgRNA, each plasmid of CD513B-cas9-U6-sgRNA (Appendix A) was transferred by Lipofectamine 3000 (Invitrogen, Carlsbad, CA, USA) into HEK293T cells. HEK293T cells were cultured in DMEM (Thermo Fisher Scientific, Waltham, MA, USA) supplemented with 10% fetal bovine serum (FBS, Thermo Fisher Scientific, Waltham, MA, USA). In brief, HEK293T cells were cultured in six-well tissue culture plates to 80% confluency and transfected with 5 μg plasmids via Lipofectamine 3000 (Invitrogen, Carlsbad, CA, USA). After 48 h, the EGFP positive cells were collected by FACS, and genomic DNA was extracted from these cell lines. For the plasmid construction of spacer sequences of pegRNA and nick sgRNA, oligos were synthesized (Appendix A), annealed, and cloned into the BbsI site of the PSK-u6 + gRNA expression vector (Appendix A). The U6-sgRNA segment was PCR-amplified using the primers listed in Appendix A. To generate the CD513B-cas9-U6-sgRNA plasmid, the amplicons were purified and constructed into the NotI site of the CD513B-cas9 vector (Appendix A).

### 4.10. Targeted Deep Sequencing of Off-Targets

To monitor possible off-target events introduced by spacer sequences of pegRNA and nick sgRNA, both potential off-target sites were predicted by Cas-OFFinder (accessed on 21 November 2021) [52]. The off-target sites were PCR-amplified from genomic DNA. First-round PCR was performed using KOD Neo (Toyobo, Shanghai, China) with the following conditions: initial denaturation at 98 °C for 1 min and then 25 cycles of denaturation at 98 °C for 10 s, annealing at 60 °C for 15 s, and extension at 68 °C for 30 s, followed by a final extension at 68 °C for 2 min. The products of first-round PCR were used as templates for the second-round PCR. Then, the second-round PCR was performed using KOD Neo (Toyobo, Shanghai, China) with the following conditions: initial denaturation at 98 °C for 1 min and then 15 cycles of denaturation at 98 °C for 10 s, annealing at 60 °C for 15 s, and extension at 68 °C for 30 s, followed by a final extension at 68 °C for 2 min. The paired-end sequencing of PCR amplicons was performed by Illumina Nextseq 500 (2 × 150) platform. Primers used for deep sequencing are listed in Appendix A.

### 4.11. Data Analysis of Deep Sequencing

The software Flash was used with default parameters to splice the fragments obtained from two terminal sequencing into a sequence. The fragments would be eliminated due to the inability to splice together to reflect the built library. The software fastx_toolkit was used to remove the low-quality sequencing fragments, and the corresponding parameters are listed as follows: fastq_quality_filter-q20-p80-v. The software cutadapt was utilized to remove the immediate sequences on both sides of the target region with the parameters are listed as follows: cutadapt-q-a-m5. We compared the sequencing fragments to the reference sequence and aligned them to determine the perfectly matched sequences. Finally, the number of perfectly matched sequences was counted to calculate the indel ratio. The parameters used for alignment: bowtie2-q-x-u-s. A self-made script carried out the counting of the alignment. The raw data of deep sequencing of off-target analysis, including each group and total number, are listed in Appendix A.

## Figures and Tables

**Figure 1 ijms-23-05948-f001:**
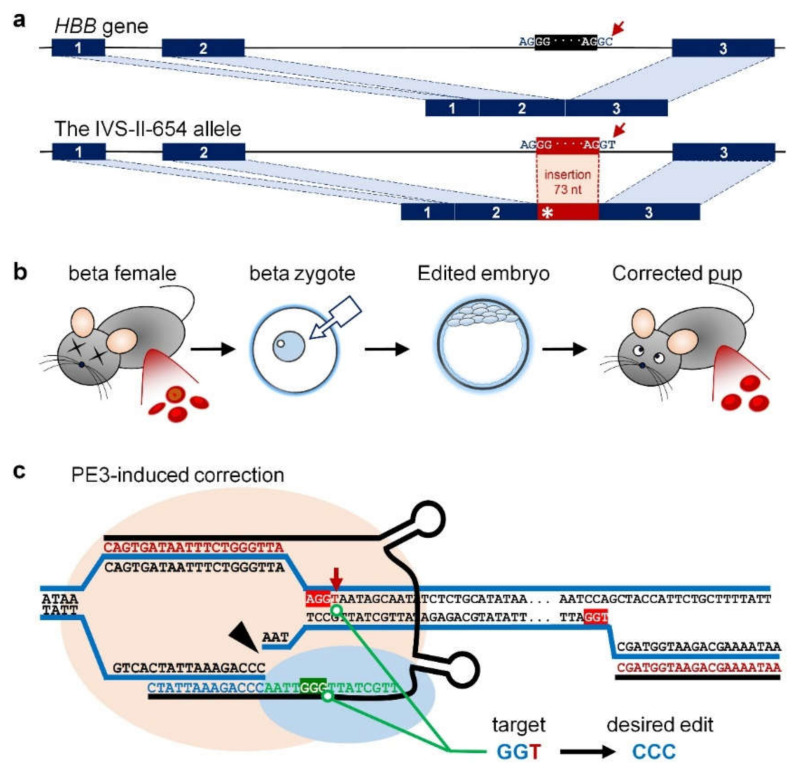
Study scheme. (**a**) Molecular mechanism of IVS-II-654 (C > T) mutation in the *HBB* gene. The IVS-II-654 mutation causes abnormal splicing, and a partial second intron sequence with a length of 73 nucleotides (nt) is thus spliced into the *HBB* mRNA, leading to a premature stop codon [4,5,6]. The asterisk indicates the premature stop codon generated by the insertion of 73-nt sequence. (**b**) Study workflow. *Hbb^th−4^*/*Hbb*^+^ (beta) mice carrying the IVS-II-654 mutation showed classic signs of beta-thalassemia, including anisocytosis and target cells in the peripheral blood. The in vitro transcripts of PE2, prime editing guide RNA (pegRNA), and nick sgRNA required for prime editor 3 (PE3) were microinjected into zygotes from beta mice. The injected zygotes developed into edited embryos, and subsequently into pups, with the corrected IVS-II-654 mutation. Beta-thalassemia symptoms in the peripheral blood were resolved by PE3-induced treatment. (**c**) PE3 used for correcting IVS-II-654 mutation in this study. The endogenous DNA sequence is shown in black. The pegRNA contained a spacer sequence and was designed to target the IVS-II-654 site (red letters). The prime binding sequence (PBS, blue letters) of pegRNA was designed to hybridize with the 3′ end of the nicked target DNA strand to form a primer-template complex. Moreover, the reverse transcriptase (RT) template (green letters) of pegRNA was designed to install a therapeutic GGT > CCC conversion at the IVS-II-654 site, which corrects the beta-thalassemia mutation and edits the SpCas9 protospacer adjacent motif (PAM) region to prevent secondary editing. The nick sgRNA (red letters in the right area) was designed to target the non-edited strand, 99-nt downstream away from the nicking position of pegRNA, for further increasing editing efficiency. Red highlight marks the PAM area; red arrow indicates the IVS-II-654 (C > T) mutation; green highlight illustrates therapeutic RT-template converting GGT to CCC. The pink shade and blue shade indicate the nCas9 domain and RT domain of PE protein, respectively. The black arrowhead indicates the nCas9 of PE protein-induced nicking position, 3-nt upstream from the PAM region.

**Figure 2 ijms-23-05948-f002:**
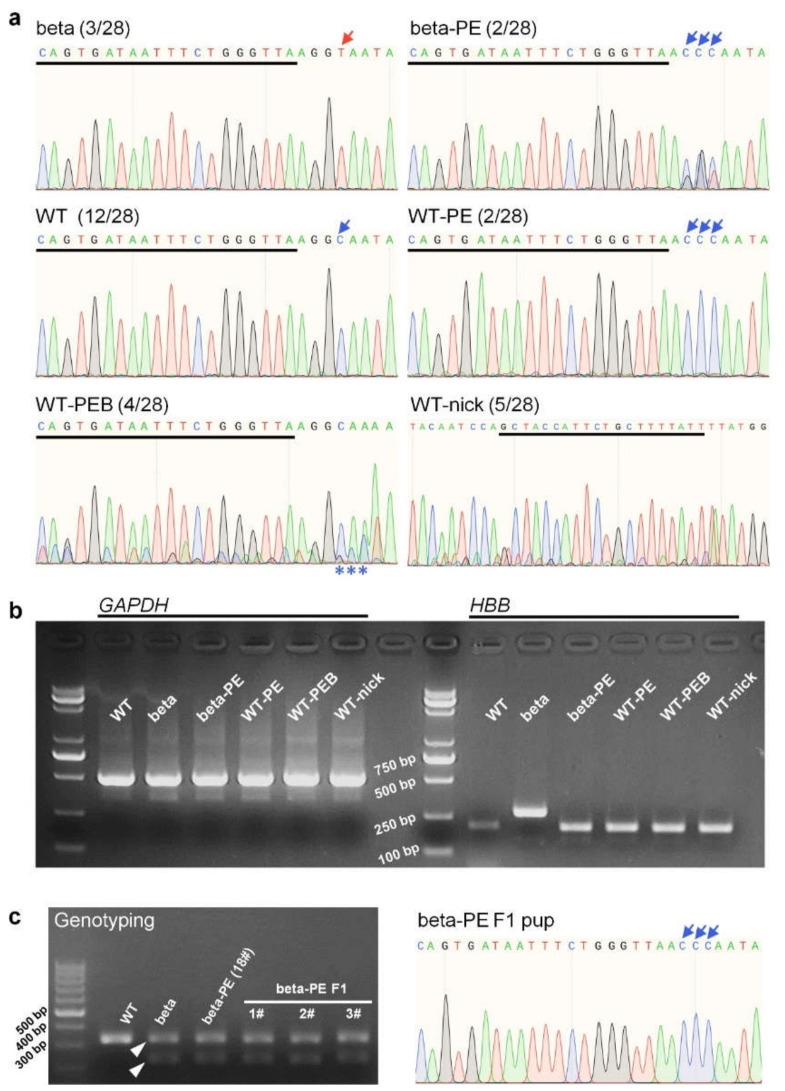
PE-editing results at the IVS-II-654 site. (**a**) Sanger sequencing traces of IVS-II-654 site. Beta and beta-PE indicate beta-genotypes (*Hbb^th−4^*/*Hbb*^+^) without and with PE3-induced editing at the IVS-II-654 site, respectively; WT and WT-PE indicate wild type (*Hbb^+^*/*Hbb*^+^) mice without and with PE3-induced editing at the IVS-II-654 site, respectively; PEB indicates prime edit with by-products; WT-nick shows WT mice containing by-product edits at the nick sgRNA site; red and blue arrows indicate mutant nucleotide (T) and WT nucleotide (C) at the IVS-II-654 site, respectively; three consecutive blue arrows indicate the desired CCC conversion; three consecutive blue asterisks indicate the desired CCC conversion in WT-PEB; black line shows the position of the spacer sequence of pegRNA, whereas the one in the traces of WT-nick indicates the position of nick sgRNA. (**b**) RT-PCR analysis of peripheral blood RNAs to detect *HBB* mRNA and *GAPDH* mRNA (as an internal control). RT-PCR of *HBB* showed that the transcript from beta mice was significantly extended because of the 73-nt sequence insertion. (**c**) Genotyping and Sanger sequencing results of IVS-II-654 mutation in the F1 generation. Electrophoresis showed that PCR-based genotyping could separate the beta and WT genotypes. The Sanger sequencing trace showed corrected editing (GGT > CCC) in the F1 beta-genotype pup.

**Figure 3 ijms-23-05948-f003:**
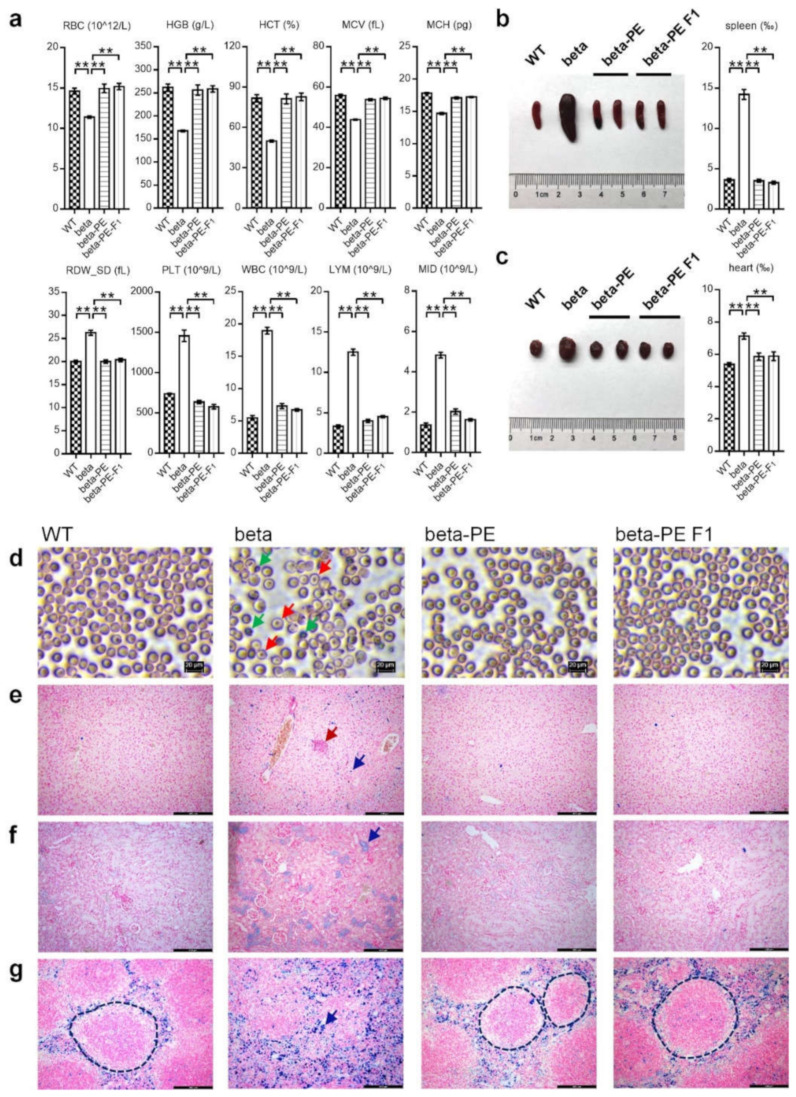
Characterization of beta-PE and beta-PE F1 mice. (**a**) Hematological analysis of red blood cell (RBC) count, hemoglobin (HGB) concentration, hematocrit (HCT), mean corpuscular volume (MCV), mean corpuscular hemoglobin (MCH), red cell distribution width_standard deviation (RDW_SD), platelets (PLT), white blood cell count (WBC), lymphocytes (LYM), and minimum inhibitory dilution (MID). Corresponding raw data are listed in Appendix A. (**b**,**c**) Analysis of spleen (**b**) and heart (**c**) coefficients, respectively. The vertical axis shows the tissue coefficient (tissue coefficient ‰ = tissue mass/body mass × 1000‰). Corresponding raw data are listed in Appendix A. Values given as mean ± standard error. ** *p* < 0.01 (*t*-test). (**d**–**g**) Microscopic examination of Wright–Giemsa-stained blood smears (**d**) and ferrocyanide iron-stained sections of the liver (**e**), kidney (**f**), and spleen (**g**). Scale bars indicate 20 µm in blood smears and 400 µm in ferrocyanide iron-stained sections. Red and green arrows indicate target cells and poikilocytosis, respectively; crimson and navy arrows indicate extramedullary hematopoiesis and iron deposition, respectively; dotted line identifies white pulp in the spleen.

**Figure 4 ijms-23-05948-f004:**
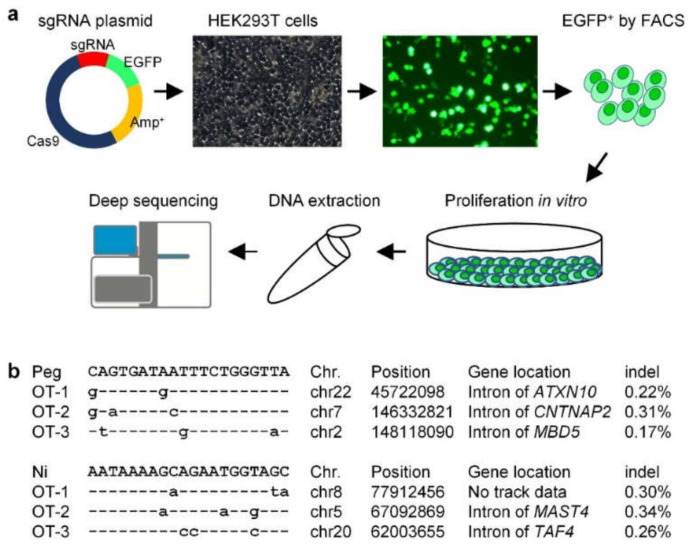
Off-target analysis of PE3 used for IVS-II-654 correction within the human genome. (**a**) Workflow of off-target analysis. Briefly, a plasmid containing Cas9, sgRNA, and the reporter gene enhanced green fluorescent protein (*EGFP*) was constructed and transferred into human HEK293T cells. *EGFP-*positive HEK293T cells were collected using fluorescence-activated cell sorting, and then proliferated in vitro to extract genomic DNA for deep sequencing of off-target edits. (**b**) According to the deep-sequencing data, the sequence, genome position, gene location, and edits (indels) of the off-targets were analyzed. Peg and Ni indicate spacer sequences of pegRNA (Peg) and nick sgRNA (Ni), respectively; OT indicates off-target.

**Table 1 ijms-23-05948-t001:** PE3-induced editing outcomes in the F0 pups.

Type	Pup	Identifier	Number	Annotation
Desired editing	beta-PE	17#, 18#	2	PE3 installed editing of GGT > CCC in beta-genotype pups
	WT-PE	21#, 22#	2	PE3 installed editing of GGC > CCC in WT-genotype pups
By-product editing	WT-PEB	9#, 28#, 30#, 32#	4	PE3 induced the CCC conversion near the IVS-II-654 site
	WT-nick	7#, 15#, 34#, 35#, 46#	5	PE3 generated by-product edits at the nick sgRNA site
Without editing	beta	20#, 24#, 31#	3	
	WT	1#, 8#, 12#, 16#, 26#, 27#, 29#, 33#, 37#, 38#, 42#, 44#	12	

## Data Availability

Not applicable.

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
