# Peer review of "Correction of Beta-Thalassemia IVS-II-654 Mutation in a Mouse Model Using Prime Editing"

_ijms, 2022, doi:10.3390/ijms23115948_

Round 1
Reviewer 1 Report
Zhang et al. have revised the manuscript and this new version explained in a more clear way their research.
All of my concerns have been addressed.
Reviewer 2 Report
Thank you for your responsiveness to reviewer comments
This manuscript is a resubmission of an earlier submission. The following is a list of the peer review reports and author responses from that submission.
Round 1
Reviewer 1 Report
The subject is relevant, and the manuscript brings important and promising information for the future treatment of beta-thalassemias. However, the manuscript needs further review to be accepted.
The last paragraph of the introduction needs to be reworded. The text is a little confusing and repetitive. Some information in this paragraph should be placed in "Materials and Methods".
In the Material and Methods, authors should include information from The ARRIVE guidelines 2.0: author checklist (https://arriveguidelines.org/resources/author-checklists).
The following items are essential to assess the reliability of the findings and reproducibility in animal experiments:
“Study design: For each experiment, provide brief details of study design including: a.The groups being compared, including control groups. If no control group has been used, the rationale should be stated.”
“Sample size: Specify the exact number of experimental units allocated to each group, and the total number in each experiment. Also indicate the total number of animals used.”
“Experimental procedures: For each experimental group, including controls, describe the procedures in enough detail to allow others to replicate them, including
- What was done, how it was done and what was used. For example, details about the embryo transfer (anesthesia, analgesia, etc.), blood collection (route, volume, anesthesia), euthanasia, and tissue collection.
- When and how often.
Author Response
Dear reviewer,
Thank you for the comments on our manuscript entitled “Correction of beta-thalassemia IVS-II-654 mutation in a mouse model using prime editing” (ID: ijms-1693090). We have studied the comments carefully and made corrections which we hope meet with approval.
Point 1: The last paragraph of the introduction needs to be reworded. The text is a little confusing and repetitive. Some information in this paragraph should be placed in "Materials and Methods".
Response 1: Thank you for this helpful comment. Accordingly, we have rewritten the last paragraph of the introduction and placed some information in 4.2.
Point 2: In the Material and Methods, authors should include information from The ARRIVE guidelines 2.0: author checklist (https://arriveguidelines.org/resources/author-checklists).
The following items are essential to assess the reliability of the findings and reproducibility in animal experiments:
“Study design: For each experiment, provide brief details of study design including: a.The groups being compared, including control groups. If no control group has been used, the rationale should be stated.”
“Sample size: Specify the exact number of experimental units allocated to each group, and the total number in each experiment. Also indicate the total number of animals used.”
“Experimental procedures: For each experimental group, including controls, describe the procedures in enough detail to allow others to replicate them, including
- What was done, how it was done and what was used. For example, details about the embryo transfer (anesthesia, analgesia, etc.), blood collection (route, volume, anesthesia), euthanasia, and tissue collection.
- When and how often.
Response 2: Thank you for your helpful comments on our article. Accordingly, we have made revisions and added brief details in the part of “Materials and Methods”.

Reviewer 2 Report
The manuscript by Zhang H et al. describes the correction of a mutation for beta-thalassemia by prime editing in the IVS-II 654 mutation. This mutation encodes for a stop codon resulting in a non-functional protein.
The authors proposed that correction with prime editing revers the aberrant splicing by microinjecting the different components in zygotes and create edited mice pups.
The strength of the present manuscript is that they achieve a 17% of edited corrected mice in which the splicing of the beta-globin mRNA is correct. This could be a first step in the correction of beta-thalassemia mutations by prime editors.
Revisions:
In general, the article is well written, well organised and explained and the English is very good.
The introduction is very concise and covers all the main points.
The results are well explained and figures helps to follow them.
The discussion concise well the article and open the possibility to how primer editors could be used in other aspect of beta-thalassemia:
Author Response
Dear reviewer,
Thank you for the comments on our manuscript entitled “Correction of beta-thalassemia IVS-II-654 mutation in a mouse model using prime editing” (ID: ijms-1693090). We have studied the comments carefully and made corrections which we hope meet with approval.
Response:Thank you for positive comments. We appreciate your efforts in reviewing our manuscript during this unprecedented and challenging time. However, the editing efficiency is miscalculated due to our little mistake. The correct efficiency is 14.29% = (2+2)/28*100%. The manuscript and the Abstract are correspondingly revised.
Reviewer 3 Report
This is a carefully performed study addressing an important clinical issue in a relevant murine model. The experiments are described in appropriate detail and experimental design is very clear. Results are clearly discussed.
1. In the abstract, the authors speak of "recovery" of hematologic parameters. This is unintentionally misleading, in that it could be taken to indicate that it reversed hematologic abnormalities in a mouse expressing them. It would be more accurate to say specifically that mice in whom the prime editor correction occurred had a hematologic phenotype similar/identical to the wild type mice as distinct from the beta mutated mice.
2. Although the authors define the various mice phenotypes/genotypes in the text, it would be helpful to the reader if they had a small table defining their terms like beta-PE, WT-PE, WT-PEB, WT-nick.
3. In 2021, a number of reviews were published which comment on prime editing for thalassemia (Lederer et al.; Brusson and Miccio; Ali et al.). At least one or two of these review should be cited.
Author Response
Dear reviewer,
Thank you for the comments on our manuscript entitled “Correction of beta-thalassemia IVS-II-654 mutation in a mouse model using prime editing” (ID: ijms-1693090). We have studied the comments carefully and made corrections which we hope meet with approval.
Point 1. In the abstract, the authors speak of "recovery" of hematologic parameters. This is unintentionally misleading, in that it could be taken to indicate that it reversed hematologic abnormalities in a mouse expressing them. It would be more accurate to say specifically that mice in whom the prime editor correction occurred had a hematologic phenotype similar/identical to the wild type mice as distinct from the beta mutated mice.
Response 1: Thank you for the suggestion. We have rewritten the corresponding part of Abstract to prevent misleading. “Subsequent comprehensive phenotypic analysis of thalassemia symptoms showed significant recovery of anemic hematological parameters, anisocytosis, splenomegaly, cardiac hypertrophy, extramedullary hematopoiesis, and iron overload following mutation correction” was changed to “Subsequent comprehensive phenotypic analysis of thalassemia symptoms, including anemic hematological parameters, anisocytosis, splenomegaly, cardiac hypertrophy, extramedullary hematopoiesis, and iron overload, showed that the corrected beta-654 mice had a normal phenotype identical to the wild-type mice.”
Point 2. Although the authors define the various mice phenotypes/genotypes in the text, it would be helpful to the reader if they had a small table defining their terms like beta-PE, WT-PE, WT-PEB, WT-nick.
Response 2: We have added the definition of beta-PE, WT-PE, WT-PEB, and WT-nick in the revised Table 1. The new Table 1 could clearly show the results of phenotypes/genotypes and editing of injected mice.
Point 3. In 2021, a number of reviews were published which comment on prime editing for thalassemia (Lederer et al.; Brusson and Miccio; Ali et al.). At least one or two of these review should be cited.
Response 3: Thank you very much for this comment. We have read these reviews carefully. According to the contents of these reviews, we cited these reviews in our manuscript.
- Koniali L, Lederer CW, Kleanthous M. Therapy Development by Genome Editing of Hematopoietic Stem Cells. Cells. 2021 Jun 14;10(6):1492. doi: 10.3390/cells10061492. PMID: 34198536; PMCID: PMC8231983. (References No. 19)
- Antoniou P, Miccio A, Brusson M. Base and Prime Editing Technologies for Blood Disorders. Front Genome Ed. 2021 Jan 28;3:618406. doi: 10.3389/fgeed.2021.618406. PMID: 34713251; PMCID: PMC8525391. (References No. 15)
- Ali S, Mumtaz S, Shakir HA, Khan M, Tahir HM, Mumtaz S, Mughal TA, Hassan A, Kazmi SAR, Sadia, Irfan M, Khan MA. Current status of beta-thalassemia and its treatment strategies. Mol Genet Genomic Med. 2021 Dec;9(12):e1788. doi: 10.1002/mgg3.1788. Epub 2021 Nov 5. PMID: 34738740; PMCID: PMC8683628. (References No. 6)
